# Causes and risk factors for an acute kidney injury outbreak among children in The Gambia, June – September 2022: A case-cohort study

Mustapha Bittaye[1], Jayne Byakika-Tusiime[2]*, Lionel Adisso[2], Boris I. Pavlin[3], Michel Muteba[2], Anna H. Jammeh[1], Ifeanyi Livinus Udenweze[4], Amadou Woury Jallow[1], Nuha Fofana[1], Momodou Kalisa[1], Sharmila Lareef[4], Kassa Mohammed Abbe[4], Patricia Eyu[4], James Nonde[2], Thierno Balde[2], Joseph Chukwudi Okeibunor[2], Otim Patrick Ramadan[2], Fiona Braka[2], Abdou Salam Gueye[2], Desta Alamerew Tiruneh[4]

**1** Ministry of Health of The Gambia, Banjul, **2** World Health Organization, AFRO, Brazzaville, Congo Republic, **3** World Health Organization, Geneva, Switzerland, **4** World Health Organization, Country Office of The Gambia, Gambia

\* jtusiime@who.int

## Abstract

### Introduction

Between June and September 2022, an outbreak of acute kidney injury (AKI) occurred in The Gambia among children, with 82 cases and 66 deaths recorded. Medicines taken by children with AKI were found to contain unacceptable amounts of diethylene glycol (DEG) and ethylene glycol (EG). The objective of the study was to establish the likely cause of the AKI outbreak and associated risk factors.

### Materials and methods

A case-cohort study was conducted in six of the seven regions of The Gambia. Cases were children eight years or younger, residing within the six regions in the study period and diagnosed with AKI within that period. Controls were children in the same age range as cases and residing within the same neighborhood as cases but without a diagnosis of AKI. Three hundred twenty-one children were recruited for the sub-cohort from which controls were selected. Data were analyzed using Marginal Structural Models for a point treatment with the inverse probability of treatment weights estimator. Multivariable logistic regression was used to identify risk factors for AKI using a 5% level of significance in the final model.

### Results

Sixty-three cases and 258 controls were enrolled into the study. Of the 63 cases, 58 were confirmed and five were suspected cases. AKI among the children was causally associated with ingestion of adulterated medicine(s) (OR = 1.76 (1.60–1.92);

**Data availability statement:** All relevant data are within the paper and its Supporting Information files.

**Funding:** This study was funded by the World Health Organization through the Contingency Fund for Emergencies The funders had no role in study design, data collection and analysis, decision to publish, or preparation of the manuscript.

**Competing interests:** The authors have declared that no competing interests exist.

$p < 0.0001$). Risk factors of AKI were the ingestion of adulterated medicine(s), concomitant medicines, and being of a younger age.

## Conclusion

The acute kidney injury outbreak that occurred among children in The Gambia in the period June through September 2022 was causally associated with the ingestion of adulterated medicines. The risk was increased by consumption of concomitant medicines and being of a younger age. DEG adulteration of paediatric medicines has been an all-too -common event in multiple countries.

## Introd uction

Acute kidney injury (AKI) refers to an abrupt decrease in kidney function that happens within a few hours or a few days [1]. The causes of AKI include obstructive uropathy, extrarenal pathology such as prerenal azotemia, specific kidney diseases such as acute interstitial nephritis or non-specific conditions such as nephrotoxicity [2]. Drug-induced nephrotoxicity is the third commonest cause of AKI [3]. Causes of AKI in children include infections such as sepsis, malaria, acute gastroenteritis, insect and snake bites, nephrotoxins such as haeme, congenital anomalies of the kidneys and post-renal causes [4–6]. The burden of AKI is very high among children in sub-Saharan Africa likely due to late presentation for treatment that leads to complications coupled with limited access to dialysis [7,8].

On 21st June 2022 Maiden Pharmaceuticals imported a consignment of pediatric syrups into The Gambia and distributed them to several pharmacies around the country for sale. Mothers/guardians of children with symptoms of cough, common cold and vomiting purchased the syrups and administered them to their children. Between 24 June and 26 July 2022, a pediatric nephrologist at Edward Francis Small Teaching Hospital (ETSTH) – the only teaching hospital in The Gambia – observed an unusual increase in the number of cases of AKI among children from five months to seven years of age, characterized mainly by fever, vomiting and oliguria or anuria [9]. All the affected children were reported to have ingested some pediatric medications for cough, cold or nausea. The suspected products were Promethazine oral solution, Kofexmalin baby cough syrup, Makoff baby cough syrup and Magrip N cold syrup [10]. The products were all manufactured by one pharmaceutical company.

The Ministry of Health (MoH-Gambia) collaborated with the World Health Organization (WHO) to test the suspected medications. Nine samples of medicines taken from children with AKI and sent for toxicological tests were found to contain unacceptable levels of diethylene glycol (DEG) and ethylene glycol (EG) [11]. No toxic substances were detected in other pediatric medicinal products supplied in The Gambia by other pharmaceutical companies [11]. Based on these results, suspected medicines were withdrawn from the community and the market. Additionally, the government of The Gambia banned the importation of the products of the incriminated pharmaceutical company and closed its operations in the country. As of 5th October 2022, 82 cases and 66 deaths had been reported. Initial epidemiological

investigations conducted by the MoH and US CDC were inconclusive as to the cause. In that study, investigators reviewed medical records and interviewed caregivers to characterize patients' symptoms and identify exposures. The preliminary investigation suggested that various contaminated syrup-based children's medications contributed to the AKI outbreak, but the actual causes of the AKI and the risk factors remained unknown [9].

An epidemiological investigation led by the WHO Regional Office for Africa (AFRO) and supported by WHO headquarters was therefore initiated at the request of the WHO Country Office (WCO) of The Gambia in support of The Gambia Government. The objective of the study was to establish the likely cause of the AKI outbreak and associated risk factors that would guide the MoH Gambia to institute appropriate control measures.

## Materials and methods

We report this study following the Strengthening the Reporting of Observational Studies in Epidemiology (STROBE) guidelines [12].

### Study design and setting

A case-cohort study design was conducted to explore the relationship between ingestion of adulterated medicines and development of AKI among children in The Gambia. The main cohort comprised all the children that resided in the villages from which the cases arose. The sub-cohort was as a fraction of the main cohort determined by proportionality to the number of confirmed cases in a given village.

This design was selected because it provides stronger evidence for causal inference than a traditional case control study and is more efficient than a cohort study design [13]. We collected primary data using interviewer-administered questionnaires and reviewed data from the national epidemiological surveillance system, line-list and preliminary epidemiological reports.

The study was conducted from 15–22 December 2022 in six of the seven health regions of The Gambia, namely Western Region 1 (WR1), Western Region 2 (WR2), Central River Region (CRR), North Bank West Region (NBWR), Lower River Region (LRR), and Upper River Region (URR) (Fig 1); there were no cases reported from North Bank East Region (NBER), thus this region was excluded.

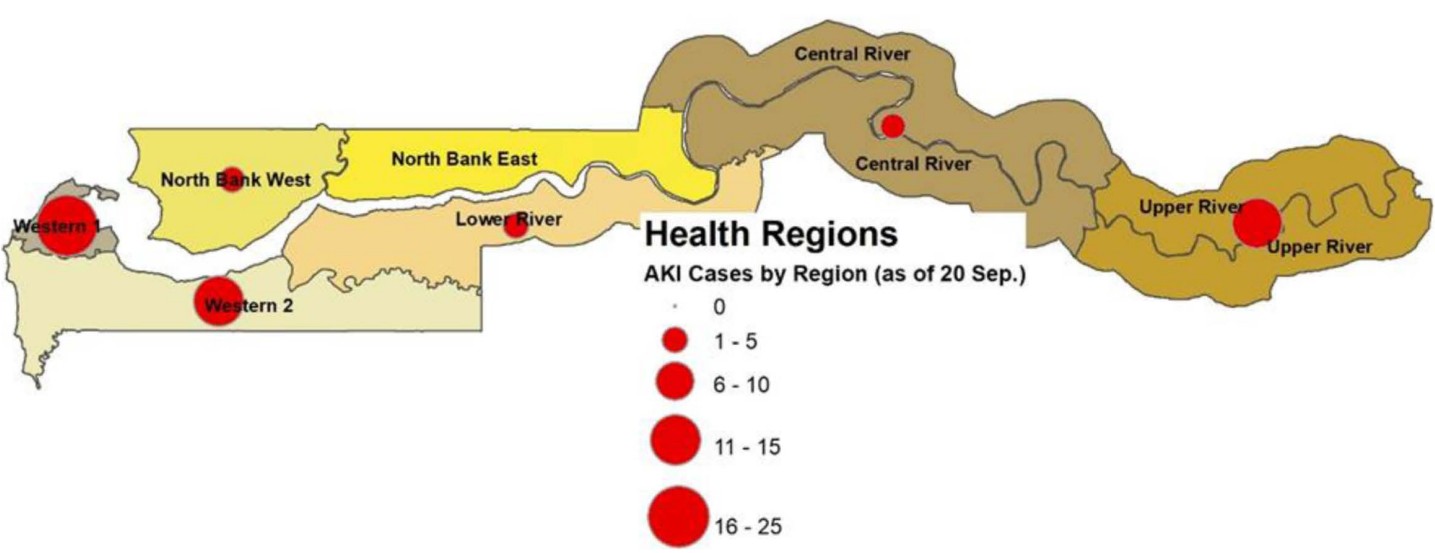

**Fig 1. Health Regions of The Gambia.**

## Participants and eligibility

The target population was all children eight years or younger in The Gambia; approximatively one third of the total population of the Gambia [14]. The source population was children residing in the six study health regions. The study population was a sample of children from each study region from whom data were collected.

Children were eligible for the study if they were born before 15th June 2022 but were not more than eight years of age and were residing within any of the six study regions in the period June to September 2022. Children were not eligible for the study if their guardians were unwilling or unable to provide informed consent.

## Cases and controls

Cases and controls for the study were selected from the main cohort which was set up from the source population that produced the cases. The case definitions used by the MoH of The Gambia and adapted for this study were:

A suspected case was a child 8 years old or younger with any of the following symptoms: fever, vomiting, diarrhea, cough, with history of syrup consumption or a child 8 years old or younger with any of the following symptoms: fever, vomiting, diarrhea, cough with reduced urine in less than 24 hours or a child 8 years old or younger with reduced urine in less than 24 hours. The definition of a suspected case was revised to enhance its sensitivity by incorporating symptoms specific to renal conditions. A probable case was any suspected case who died without confirmation. A confirmed case was any suspected case with acute onset of either oliguria or anuria of unknown etiology lasting for more than 24 hours or a suspected case confirmed with serum creatinine levels above normal, that is measured creatinine greater than 1.5–2 times the reference creatinine, with unknown etiology. In our study, a case was any child diagnosed with AKI within the period June to September 2022.

A control was defined as a child eight years or younger with no AKI diagnosis and residing within the same village as a case. Controls were not matched to cases.

## Sample size

A sample size of 500 children was required to achieve 80% power to detect a minimum clinically significant difference of 15% in AKI at a one tailed 5% level of statistical significance and an Intra Correlation Coefficient of 0.05. The sample size was calculated using the SAS statistical software version 9.4. [15] In addition, assuming a non-response rate of 20%, and accounting for controls who may become cases, a total sample size of 625 children was determined for the main cohort.

Three hundred thirty-six children were selected for the sub-cohort from which controls were identified. The sub-cohort was a fraction of the main cohort determined proportionally to the number of cases [16].

## Sampling procedure

We used cases on the line – list. These cases were identified from records of all hospitals in the country. All 82 confirmed cases were sought for inclusion in the study. We also sought to include all suspected or probable cases from WR1 and WR2 that produced the highest number of cases. Suspected and probable cases from other regions were not included due to financial constraints. Controls were selected using a simple random sampling technique. All the 49 villages from which cases arose were included in the study. In the villages, the number of children in the main cohort and the sub cohort was predetermined using probability proportionate to size technique. In these villages, using the Compass app®, we identified randomly households where controls would be selected according to a predetermined sample size. A child was then selected randomly from the selected household. Three controls were selected for each case.

## Training of the research team

Research assistants fluent in the respective local languages of the study regions, were recruited and trained in effective data collection methods.

## Recruitment of study participants

With the help of the clinicians at the hospital, children who were confirmed with AKI were identified from the hospital register. The research team comprised surveillance officers that had interacted with the children previously when the outbreak was first identified. Members of the research team approached the mothers or guardians of the children at their homes, introduced the study to them and asked if they were interested in participating in the study. The research staff explained, in detail, the purpose of the study and the procedures to the mothers/guardians of the children (dead or alive) and then asked for their written consent. Children were enrolled into the study if their mothers/guardians consented to the study.

Controls were randomly selected from the sub cohort. Parents or guardians of selected children were approached by the research staff to seek their consent for their children to participate in the study. Those that consented and were eligible were enrolled into the study.

## Data collection and measurement

Data were collected using an interviewer administered online questionnaire on Kobo Collect application® that was pre-tested on two cases and six controls. The questionnaire was pre-tested in two of the study regions and revised according to the findings from the pretest. We collected data on sociodemographic characteristics; primary water source for drinking; electricity availability; availability of some household equipment assets such as television; refrigerator; common paediatric risk factors for AKI, including *E. Coli* haemolytic uremic syndrome (HUS), particularly because of the concern related to deteriorations in water, sanitation and hygiene (WASH) associated with antecedent flooding; and clinical characteristics of participants. The full questionnaire can be provided on request.

## Statistical analyses

We defined two new variables "income" and "income level". The variable "income" was created from the following collected variables: "primary water source for drinking", "does the household have electricity?", "does the household have a television?" and "does the household have a refrigerator?". Those variables used were binary (0,1) except the "primary water source for drinking" which was a categorical variable with eleven categories recorded into an ordinal variable with 3 level (1,2, 3). The binary variables were recoded 1 for Yes and 0 for No. The variable "income" was determined as a composite score from the other four variables We considered the variable "income" as a proxy of the actual income of the household that we were not able to measure. The variable "income" was an ordinal variable from 0 to 6. The highest score represented the highest income. The income score was then categorized into levels of income under the variable "income level" as follows: income <3 for *low income*; 3 ≤ income ≤ 4 for *middle income* and income >4 for *high income*. For the analytic analyses, were created a binary variable "caregiver" to determine if one was the mother of the child or not.

Descriptive data analyses by group were performed on the participants' characteristics using frequencies and percentages or medians and interval quartile range (IQR), as appropriate.

Marginal Structural Models (MSMs) for point treatment with an inverse probability of treatment weights [IPTW] estimator were employed to establish the causal association between drug ingestion and development of AKI among the children [17]. A series of treatment models were tested and the Akaike information criteria (AIC) was used to identify the best treatment model [18]. The primary outcome was the development of AKI. The primary exposure was the ingestion by the children of any of the following medicines: (*Promethazine oral solution BP*, *Kofexmalin baby cough syrup*, *Makoff baby cough syrup*, or *Magrip N cold syrup*). Other important exposures examined were co-infections, consumption of concomitant drugs, natural toxins from plants, animals, and industrial chemicals.

To identify risk factors for AKI, logistic regressions were used [19]. First, potential risk factors were selected using univariate logistic regression with a conservative p-value ≤ 0.20. Second, a multivariable analysis using a backward stepwise approach was conducted with all the selected variables from the previous stage. Risk factors of AKI were the variables with p-value ≤ 0.05 in the final model [19].

## Ethical approval

Ethical approval to conduct the study was obtained from WHO AFRO's Ethics Review Committee (Protocol ID: AFR/ERC/2022/12.4) and the University of The Gambia Ethics Committee (UTGEC 01.12.22). Informed consent was obtained from mothers/guardians.

## Results

### Participants

Sixty-three cases and 258 controls were enrolled into the study. No control became a case. Of the cases, 58 were confirmed and five were suspected (of the total 11 suspected cases). (Fig 2).

### Participant characteristics

**Demographic characteristics.** Cases were generally younger than controls (median = 19 months; IQR = 12–30 vs median = 33 months; IQR = 20–52) respectively. There were more males among cases than among controls (61.9% vs 49.6%) respectively. Other sociodemographic characteristics were balanced between the cases and controls. Participants were mostly Gambian, Mandinka and of urban residence in west coast regions (WR1, WR2). Most participants belonged to families within the high-income bracket as per Gambian standards. That is, they had piped water in their dwellings, had electricity, television, and a refrigerator. Quranic education was the commonest level of education attainment for guardians. (Table 1).

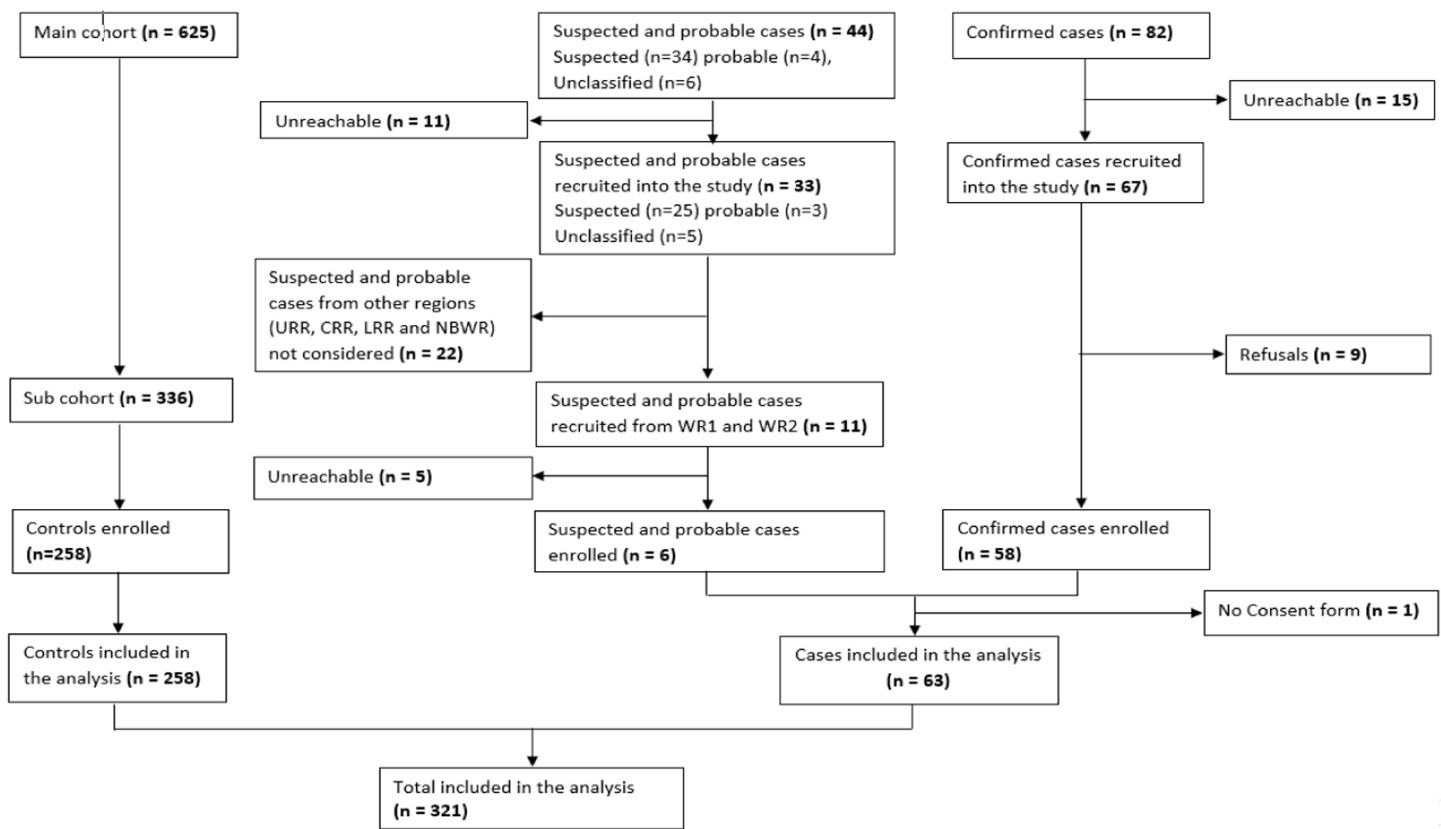

**Fig 2. Study Flow Chart.**

**Table 1. Socio demographic characteristics.**

| Socio-demographic characteristics | N | No AKI | AKI |
|---|---|---|---|
| **Age**, n, median (IQR), months | 321 | 258, **33**.0 (20–52) | 63, **19** (12-30) |
| **Sex**, males, n (%) | 321 | 128 (49.6) | 39 (61.9) |
| **Height**, median (IQR), cm | 258 | 240, 73 (38–93) | 17, 65 (34–81) |
| **Weight**, median (IQR), Kg | 257 | 240, 12.4(9.7–15.7) | 17, 10.0(7.4–12.1) |
| **Mid-upper arm circumference (MUAC)**, median (IQR), cm | 249 | 232, 14.9(13.5-15.8) | 17, 14.0(13.2–15.2) |
| **Nationality, n (%)** | 321 | 258 | 63 |
| Gambian only | | 244 (94.7) | 61 (96.8) |
| Dual Citizen | | 3 (1.1) | 1 (1.6) |
| Non-Gambian | | 8 (3.1) | 1 (1.6) |
| Missing | | 3 (1.1) | 0 (0.0) |
| **Ethnic group, n (%)** | 321 | 258 | 63 |
| Mandinka | | 87 (33.7) | 25 (39.7) |
| Fula | | 47 (18.2) | 15 (23.8) |
| Sarahuleh | | 38 (14.7) | 12 (19.0) |
| Wollof | | 45 (17.4) | 2 (3.2) |
| Jola | | 17 (6.6) | 4 (6.4) |
| Other | | 24 (9.3) | 5 (7.9) |
| **Region of residence, n (%)** | 321 | 258 | 63 |
| Western region 1 (WR1) | | 131 (50.8 | 34 (54.0) |
| Western region 2 (WR2) | | 64 (24.8) | 12 (19.0) |
| Upper river region (URR) | | 45 (17.4) | 12 (19.0) |
| Central river region (CRR) | | 9 (3.5) | 2 (3.2) |
| North bank west region (NBWR) | | 6 (2.3) | 2 (3.2) |
| Lower river region (LRR) | | 3 (1.2) | 1 (1.6) |
| **Area of residence, n (%)** | 321 | 258 | 63 |
| Urban | | 197 (76.4) | 46 (73.0) |
| Rural | | 61 (23.7) | 17 (27.0) |
| **Flooding in area of residence, n (%)** | 321 | 258 | 63 |
| Yes | | 59 (22.9) | 6 (9.5) |
| No | | 199 (77.1) | 55 (87.3) |
| Missing | | 0 (0.0) | 2 (3.2) |
| **Income level, n (%)** | 321 | 258 | 63 |
| High | | 186 (72.1) | 47 (74.6) |
| Middle | | 42 (16.3) | 9 (14.3) |
| Low | | 30 (11.6) | 7 (11.1) |
| **Relation caregiver – child, n (%)** | 321 | 258 | 63 |
| Mother | | 214 (83.0) | 44 (69.8) |
| Father | | 19 (7.4) | 17 (27.0) |
| Grandparent | | 11 (4.3) | 1 (1.6) |
| Other relatives | | 9 (3.5) | 1 (1.6) |
| Non-relatives | | 5 (1.9) | 0 (0.0) |
| Caregiver (mother), n (%) | 321 | 214 (83.0) | 44 (69.8) |
| **Education of the child's mother, n (%)** | 321 | 258 | 63 |
| Tertiary | | 21 (8.2) | 4 (6.2) |
| Secondary | | 69 (26.7) | 15 (23.4) |
| Primary | | 26 (10.1) | 9 (14.1) |

*(Continued)*

**Table 1.** (Continued)

| Socio-demographic characteristics | N | No AKI | AKI |
|---|---|---|---|
| None | | 46 (17.8) | 8 (12.7) |
| Quranic only | | 69 (26.7) | 20 (31.7) |
| Vocational | | 0 (0.0) | 1 (1.6) |
| Missing | | 27 (10.5) | 6 (9.5) |
| **Education of the child's father, n (%)** | 321 | 258 | 63 |
| Tertiary | | 22 (8.5) | 0 (0.0) |
| Secondary | | 18 (7.0) | 5 (7.9) |
| Primary | | 6 (2.3) | 1 (1.6) |
| None | | 9 (3.5) | 1 (1.6) |
| Quranic only | | 45 (17.4) | 11 (17.5) |
| Vocational | | 2 (0.8) | 0 (0.0) |
| Missing | | 156 (60.5) | 46 (71.4) |
| **Household head, n (%)** | 321 | 258 | 63 |
| Father | | 156 (60.5) | 45 (71.4) |
| Mother | | 25 (9.7) | 6 (9.5) |
| Grandparent | | 52 (20.1) | 9 (14.3) |
| Other relatives | | 2 (0.8) | 0 (0.0) |
| Non-relatives | | 23 (8.9) | 3 (4.8) |
| **Household head's main occupation, n (%)** | 321 | 258 | 63 |
| Labourer | | 62 (24.0) | 20 (31.7) |
| Peasant | | 44 (17.1) | 9 (14.3) |
| Professional | | 40 (15.1) | 8 (12.7) |
| Trader | | 76 (29.9) | 16 (25.4) |
| Other | | 29 (11.2) | 7 (11.1) |
| Missing | | 7 (2.7) | 3 (4.8) |

**Clinical characteristics.** Fever (n = 42; 66.7%), vomiting (n = 37; 58.7%), loss of appetite (n = 30; 47.6%), anuria (n = 49; 77.8%), wheezing (n = 11; 17.5%) and diarrhea (n = 20; 31.7%) were more prevalent among cases than controls. Cases consumed more medicines of all types than controls: antibiotics (17.5% vs 8.7%) concomitant drugs (antibiotics, anti-inflammatory drugs and antimalaria drugs) (58.7% vs 6.2%) and suspected adulterated medicines (58.7% vs 6.2%) (Table 2).

## Outcomes

**Cause of AKI.** The results of the MSM analysis show that AKI among the children was causally associated with ingestion of adulterated medicine (OR = 1.76 (1.60; 1.92); p < .0001). That is, children who ingested adulterated medicines had almost twice the odds of developing AKI compared to children who did not ingest adulterated medicines.

**Risk factors for AKI among children in The Gambia.** The multivariable logistic regression results show that risk factors for AKI were the ingestion of adulterated medicines, concomitant drugs, and younger age. Between two Gambian children with a one-year difference, who were equally exposed to adulterated medicines and concomitant drugs, the younger child had 3% higher odds of developing AKI compared to the older child (OR = 1.03 (CI 95% 1.01 to 1.05). Between two Gambian children of the same age, and equally exposed to concomitant drugs, the one exposed to adulterated medicines had much higher odds of developing AKI (OR = 9.70 (CI 95% 4.25 to 23.10). Between two

**Table 2. Clinical characteristics.**

| Clinical characteristics | N | No AKI | AKI |
|---|---|---|---|
| **Comorbidity (yes), n (%)** | | | |
| **Fever** | 321 | 258 | 63 |
| Yes | | 86 (33.3) | 42 (66.7) |
| No | | 164 (63.6) | 21 (33.3) |
| Missing | | 8 (3.1) | 0 (0.0) |
| **Cough** | 321 | 258 | 63 |
| Yes | | 77 (29.8) | 19 (30.2) |
| No | | 179 (69.4) | 44 (69.8) |
| Missing | | 2 (0.8) | 0 (0.0) |
| **Vomiting** | 321 | 258 | 63 |
| Yes | | 34 (13.2) | 37 (58.7) |
| No | | 221 (85.7) | 26 (41.3) |
| Missing | | 3 (1.1) | 0 (0.0) |
| **Loss of appetite** | 321 | 258 | 63 |
| Yes | | 31 (12.0) | 30 (47.6) |
| No | | 223 (86.4) | 32 (50.8) |
| Missing | | 4 (1.6) | 1 (1.6) |
| **Runny nose** | 321 | 258 | 63 |
| Yes | | 44 (17.1) | 10 (15.9) |
| No | | 211 (81.8) | 53 (84.1) |
| Missing | | 3 (1.1) | 0 (0.0) |
| **Anuria** | 321 | 258 | 63 |
| Yes | | 0 (0.0) | 49 (77.8) |
| No | | 255 (81.8) | 14 (22.2) |
| Missing | | 3 (1.1) | 0 (0.0) |
| **Abdominal pain** | 321 | 258 | 63 |
| Yes | | 27 (10.5) | 17 (29.3) |
| No | | 227 (88.0) | 41 (65.8) |
| Missing | | 4 (1.5) | 5 (7.9) |
| **Chest pain** | 321 | 258 | 63 |
| Yes | | 34 (13.2) | 8 (12.7) |
| No | | 221 (85.7) | 49 (77.8) |
| Missing | | 3 (1.1) | 6 (9.5) |
| **Rash** | 321 | 258 | 63 |
| Yes | | 27 (10.4) | 2 (3.2) |
| No | | 228 (88.4) | 61 (96.8) |
| Missing | | 3 (1.2) | 0 (0.0) |
| **Diagnosis of malaria** | 321 | 258 | 63 |
| Yes | | 19 (7.4) | 4 (6.3) |
| No | | 237 (91.8) | 59 (93.7) |
| Missing | | 2 (0.8) | 0 (0.0) |
| **Wheezing** | 321 | 258 | 63 |
| Yes | | 10 (3.9) | 11 (17.5) |
| No | | 247 (95.7) | 50 (79.4) |
| Missing | | 1 (0.4) | 2 (3.1) |
| **Diarrhea** | 321 | 258 | 63 |

*(Continued)*

| Clinical characteristics | N | No AKI | AKI |
|---|---|---|---|
| Yes | | 29 (11.2) | 20 (31.7) |
| No | | 223 (86.5) | 41 (65.1) |
| Missing | | 6 (2.3) | 2 (3.2) |
| **Total number of unique medications given (0, 1, 2 or more), n (%)** | 321 | 258 | 63 |
| 0 | | 141 (54.7) | 10 (15.9) |
| 1 | | 62 (24.0) | 13 (20.6) |
| 2 | | 33 (12.8) | 10 (15.9) |
| ≥3 | | 22 (8.5) | 30 (47.6) |
| **Antibiotics taken (≥1), n (%)** | 317 | 21 (8.7) | 11 (17.5) |
| **Anti-inflammatory given (≥1), n (%)** | 311 | 47 (18.7) | 39 (65.0) |
| **Consumption of acetaminophen, n (%)** | 321 | 258 | 63 |
| Yes | | 36 (14.0) | 35 (55.6) |
| No | | 186 (72.0) | 15 (23.8) |
| Missing | | 36 (14.0) | 13 (20.6) |
| **Suspected adulterated medicines** | 321 | 258 | 63 |
| Yes | | 16 (6.2) | 37 (58.7) |
| No | | 238 (92.2) | 25 (39.7) |
| Missing | | 4 (1.6) | 1 (1.6) |
| **Promethazine oral solution BP, n (%)** | 321 | 258 | 63 |
| Yes | | 10 (3.9) | 31 (49.2) |
| No | | 243 (94.2) | 29 (46.0) |
| Missing | | 5 (1.9) | 3 (4.8) |
| **Kofexmalin baby cough syrup ingested, n (%)** | 321 | 258 | 63 |
| Yes | | 4 (1.5) | 12 (19.0) |
| No | | 252 (97.7) | 50 (79.4) |
| Missing | | 2 (0.8) | 1 (1.6) |
| **Makoff baby cough syrup ingested, n (%)** | 321 | 258 | 63 |
| Yes | | 3 (1.1) | 9 (14.3) |
| No | | 253 (98.1) | 53 (84.1) |
| Missing | | 2 (0.8) | 1 (1.6) |
| **Magrip N cold syrup, ingested, n (%)** | 321 | 258 | 63 |
| Yes | | 1 (0.4) | 5 (7.9) |
| No | | 254 (98.4) | 57 (90.5) |
| Missing | | 3 (1.2) | 1 (1.6) |
| **Concomitant drugs (≥1), n (%)** | 275 | 57 (22.2) | 41 (64.1) |

Gambian children of the same age, and equally exposed to adulterated medicine, the one exposed to concomitant drugs had higher odds of developing AKI (OR = 3.28 (CI 95% 1.45 to 7.53) (Table 3).

## Discussion

This study evaluated if the development of AKI among children in The Gambia from June through September 2022 [9–11] was causally associated with ingestion of adulterated medicines. We observed higher odds of developing AKI among children who ingested the specified adulterated medicines (*Promethazine* oral solution BP, *Kofexmalin* baby cough syrup, *Makoff* baby cough syrup, *Magrip* N cold syrup) compared to children who did not. The temporal relationship between the

**Table 3. Risk factors for AKI among Gambian children.**

| Variables | aOR | CI 95% | P value |
|---|---|---|---|
| Age | 1.03 | 1.01–1.05 | **.009** |
| Sex | 1.69 | 0.78–3.74 | .182 |
| Caregiver | 0.42 | 0.17–1.09 | .071 |
| Exposure to adulterated medicines | 9.70 | 4.25–23.10 | **<.001** |
| Exposure to concomitants drugs | 3.28 | 1.45–7.53 | **.003** |

importation of the suspected medicines into the country, their consumption and development of AKI among the children is supportive of this hypothesis. This was further supported by the findings of the causality assessment [11].

High levels of diethylene glycol (DEG) and ethylene glycol (EG) were detected in drug samples that were sent to laboratories in Ghana, France, and Switzerland [20]. Intentional or accidental ingestion of DEG/EG-containing products causes toxicity and may be lethal. The kidney injury in DEG poisoning is secondary to proximal tubular necrosis caused by diglycolic acid (DGA). Marked vacuolization and edema of epithelial cells obstruct the lumen, reducing urine flow and, consequently, resulting in anuria and uremia. The clinical alterations due to DEG poisoning are dose-dependent. Patients may present with gastrointestinal symptoms and anion gap metabolic acidosis, followed by renal failure [21]. Our results are consistent with findings and conclusions of the causality assessment conducted by MoH – The Gambia, on acute kidney injury in children [11]. The rapid reduction in the number of new cases following the recall of the medicines provides further evidence of the association.

The alternative explanation of *E. coli* infection is unlikely because only 8% of cases presented with bloody diarrhea which increases the risk of developing serious complications such as acute renal failure. Only two samples tested positive for Shiga toxin-producing *E. coli*.

There have been several epidemics of acute renal failure affecting predominantly young children where the cause has been DEG poisoning; many children worldwide have died from DEG poisoning [22]. DEG is a toxic alcohol used in brake fluid, paint, and household cleaning products, and has been used illegally as a cheap substitute solvent in place of glycerol in drug manufacturing. Previous DEG poisonings resulting from contamination of medications have been reported in the United States, Nigeria (1990), Panama, and other countries, and acute renal failure is a known manifestation of DEG poisoning [23,24]. The poisoning was the result of either contamination of the medicinal products by DEG or the deliberate illegal use of DEG as a solvent. Between June and September 1990, 47 children died at Jos University Teaching Hospital, Nigeria from ingestion of paracetamol syrup adulterated with diethylene glycol. Most of the children presented with anuria, fever, vomiting, diarrhea, and convulsions[8]. All died within two weeks of admission. A similar event happened in Port-au-Prince, Haiti between November 1995, and May 1996 where 109 cases of acute renal failure among children were identified. Of 87 patients with follow up information who remained in Haiti for treatment, 85 (98%) died. A locally manufactured acetaminophen syrup was highly associated with the outbreak (OR 52.7 95% CI 15.2–197.2) DEG was found in patients' bottles in a medicine concentration of 14.4%. Glycerin, a raw material imported to Haiti and used in the acetaminophen formulation, was contaminated with 24% DEG [24]. In the short time since this investigation, DEG/EG adulteration of paediatric medicines has been identified in at least six countries: Indonesia (WHO 2022a), Uzbekistan and Cambodia (WHO 2023a), Iraq (WHO 2023b), the Marshall Islands and the Federated States of Micronesia (WHO 2023c), and Cameroon (WHO 2023d); with the medicines or excipients originating in India and China. Health professionals need to be aware of the clinical presentation of DEG poisoning and report it to public health authorities immediately, as prompt action is likely to save lives by the removal of the contaminated/illegal medicine from pharmacies and shops in the affected area.

From this study, risk factors for AKI among children in The Gambia were consumption of adulterated medicines, concomitant drugs (antibiotics and anti-inflammatory drugs), and being of a younger age. These factors have been

documented in other studies [25]. Many of the children who were given the adulterated medicines had no indication for their use – such syrups are not recommended for children under two years of age – yet this practice is extremely common in The Gambia.

## Study limitations

Like all case control studies where participants must recall past events, there is always a risk of forgetting and recall bias: while cases tend to remember everything that happened to them and more, the reverse is true for controls, hence the bias. To prod participants' memories, a popular feast, *Tobaski*, was used as a reference date. The research team was trained to equally probe for experiences in both cases and controls significantly reducing recall bias that may exaggerate the effect among the cases (bias away from the null). Some guardians declined to participate in the study because they probably did not want to be reminded of the loss of their children. This was a potential source of selection bias; however, only 13% of cases declined to participate. If the cases that declined to participate all had consumed the adulterated medicines, this would reduce the size of the effect observed.

The findings of this study contribute to the body of knowledge on causes of AKI among children but are not conclusive on their own. We cannot infer causality from only one study. Other pieces of evidence need to be identified and together with the findings from this study conclude on the cause of AKI among children in The Gambia. That said, however, we can support our conclusions with what is already known on this topic of "DEG/EG causing AKI" basing on the fact that DEG and EG were found in the samples from the victims that were tested. Employing three necessary conditions for causality – covariation, temporal precedence and ruling out rival explanations [26] – at least two of the criteria can be supported by the available data. Covariation was demonstrated in the findings where close to 60% of children that consumed suspected medicines developed AKI while 92% of children who did not consume suspected medicines did not develop AKI. Some questions remain because 40% of children with AKI did not ingest suspected medicines and 8% of children who did not ingest suspected medicines also developed AKI. AKI has various etiologies, including specific kidney diseases (e.g., acute interstitial nephritis, acute glomerular and vasculitic renal diseases); non-specific conditions (e.g., ischemia, toxic injury); as well as extrarenal pathology (e.g., prerenal azotemia, and acute postrenal obstructive nephropathy) that can co-exist and complicate recognition and treatment. In this study we did not perform clinical and laboratory investigations to rule out alternative causes because close to 90% of cases were deceased at the time of the investigation, and also because sophisticated diagnostic capacity is not available in The Gambia (for example, routine blood chemistry was only made available after this incident). Temporal precedence was fulfilled as the records clearly indicated that the AKI diagnosis was made after the victims had ingested the suspected medicines. In addition, this outbreak occurred after the importation of the suspected medicines into the country according to reports from the MoH and was terminated by withdrawal of the medicines. The third criterion of ruling out rival explanations may be linked to the discussion under the first criterion. Not all possible rival explanations were excluded. Nevertheless, findings of a causality assessment of adverse drug reaction conducted at the same time by a team of experts corroborate the findings of our study.[10] In this review, the team of experts determined that 56 cases of AKI and 22 deaths were due to medicines adulterated with DEG/EG.[10] Like our findings, the team of experts noted that 34 cases did not have confirmed exposure to the suspected medicines. However, it is likely that at least some of these cases were due to other common causes of paediatric AKI identified as a result of heightened surveillance during this outbreak; others may have received the implicated medicines but did not report them for whatever reason.

## Conclusions

AKI among children in The Gambia was causally associated with ingestion of medicines adulterated with DEG/EG. This was compounded by concurrent use of other medicines. Because of the grey areas in our findings and those of the causality assessment of drug reaction, more evidence is needed to confirm without doubt that the AKI outbreak among

children in The Gambia from June through September 2022 was fully attributable to medicines adulterated with DEG/EG. Current evidence suggests that at least 60% of cases were due to medicines adulterated with DEG/EG; the remaining cases were likely due to other common causes of AKI identified as a result of enhanced surveillance during this outbreak.

## Recommendations

The government of The Gambia should strengthen its regulatory oversight of the pharmaceutical industry from importation to distribution of medicines and implement strict quality control measures for imported medicines through inspection mechanisms including testing of samples for better pharmacovigilance and toxicovigilance to prevent importation of and improve detection of sub-standard medicines and intoxication with adulterated medicines.

Health providers and the public should be sensitized about the dangers of adulterated medicines, their identification and reporting to competent health authorities. Public awareness should be conducted as well about irrational medication use and polypharmacy, especially for young children, which are risk factors for adverse drug reactions such as nephrotoxicity.

Collaboration with expert institutions such as the WHO and the International Pharmaceutical Federation (FIP) should be encouraged to access resources and expertise in pharmaceutical quality control.

The Gambian government should enforce strict penalties for individuals and organizations found to be involved in the importation, distribution, or sale of adulterated medicines.

Local production of essential medicines to reduce reliance on imported medicines should be encouraged to improve the safety and efficacy of medicines available to young children and protect them from the risks associated with adulterated medicines.

Engage in regional cooperation and information sharing to address cross-border issues related to medicine importation and quality control.

To improve healthcare in Africa, it is recommended to increase advocacy efforts and enhance the provision of organ support services, with a particular focus on expanding pediatric dialysis facilities.

## Supporting information

**S1. Database_AKI_Study_Dec22_Anonymized.**
(XLSX)

## Acknowledgments

We wish to acknowledge the goodwill of all study participants (children and guardians, dead and alive) that enabled us to conduct this study. We thank all members of the study team that worked tirelessly to collect the data amidst a hostile environment of depression and anger from bereaved parents. We thank the MoH of The Gambia and its leadership for the incredible support they provided for this study.

## Author contributions

**Conceptualization:** Mustapha Bittaye, Jayne Byakika-Tusiime, Boris I Pavlin, Thierno Balde, Fiona Braka, Desta Alamerew Tiruneh, Abdou Salam Gueye.

**Data curation:** Mustapha Bittaye, Jayne Byakika-Tusiime, Lionel Adisso, Boris I Pavlin, Michel Muteba, Anna H. Jammeh, Ifeanyi Livinus Udenweze, Amadou Woury Jallow, Nuha Fofana, Momodou Kalisa, Sharmila Lareef, Kassa Mohammed Abbe, Patricia Eyu, James Nonde, Thierno Balde, Joseph Chukwudi OKEIBUNOR, Fiona Braka.

**Formal analysis:** Mustapha Bittaye, Jayne Byakika-Tusiime, Lionel Adisso, Boris I Pavlin, Michel Muteba, Anna H. Jammeh, Ifeanyi Livinus Udenweze, Momodou Kalisa, Sharmila Lareef, Kassa Mohammed Abbe, Patricia Eyu, James Nonde, Thierno Balde, Joseph Chukwudi OKEIBUNOR, Fiona Braka.

**Investigation:** Jayne Byakika-Tusiime, Lionel Adisso, Boris I Pavlin, Michel Muteba, Anna H. Jammeh, Ifeanyi Livinus Udenweze, Amadou Woury Jallow, Nuha Fofana, Momodou Kalisa, Sharmila Lareef, Kassa Mohammed Abbe, Patricia Eyu, James Nonde, Fiona Braka.

**Methodology:** Jayne Byakika-Tusiime, Lionel Adisso, Boris I Pavlin, Michel Muteba, Anna H. Jammeh, Ifeanyi Livinus Udenweze, Amadou Woury Jallow, Nuha Fofana, Momodou Kalisa, Sharmila Lareef, Kassa Mohammed Abbe, Patricia Eyu, James Nonde, Joseph Chukwudi OKEIBUNOR, Fiona Braka.

**Project administration:** Amadou Woury Jallow, Thierno Balde, Joseph Chukwudi OKEIBUNOR, Desta Alamerew Tiruneh.

**Resources:** Fiona Braka, Abdou Salam Gueye.

**Supervision:** Thierno Balde, Fiona Braka, Desta Alamerew Tiruneh, Abdou Salam Gueye.

**Visualization:** Lionel Adisso.

**Writing – original draft:** Mustapha Bittaye, Jayne Byakika-Tusiime, Lionel Adisso, Boris I Pavlin, Michel Muteba, Anna H. Jammeh, Ifeanyi Livinus Udenweze, Amadou Woury Jallow, Nuha Fofana, Momodou Kalisa, Sharmila Lareef, Kassa Mohammed Abbe, Patricia Eyu, James Nonde, Thierno Balde, Joseph Chukwudi OKEIBUNOR, Fiona Braka, Desta Alamerew Tiruneh, Abdou Salam Gueye.

**Writing – review & editing:** Mustapha Bittaye, Jayne Byakika-Tusiime, Lionel Adisso, Boris I Pavlin, Michel Muteba, Anna H. Jammeh, Ifeanyi Livinus Udenweze, Amadou Woury Jallow, Nuha Fofana, Momodou Kalisa, Sharmila Lareef, Kassa Mohammed Abbe, Patricia Eyu, James Nonde, Thierno Balde, Joseph Chukwudi OKEIBUNOR, Fiona Braka, Desta Alamerew Tiruneh, Abdou Salam Gueye.

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
