## [Decision Letter · Decision Letter 0]

PONE-D-24-22500Causes and risk factors for an acute kidney injury outbreak among children in The Gambia, June – September 2022: A case-cohort studyPLOS ONE

Dear Dr. Office,

Thank you for submitting your manuscript to PLOS ONE. After careful consideration, we feel that it has merit but does not fully meet PLOS ONE’s publication criteria as it currently stands. Therefore, we invite you to submit a revised version of the manuscript that addresses the points raised during the review process. Please observe all editor and reviewer comments and give a point-to-point response to every comment they provided.

We look forward to receiving your revised manuscript.

Kind regards,

Tungki Pratama Umar, M.D.

Academic Editor

PLOS ONE

   "This study was funded by the World Health Organization through the Contingency Fund for Emergencies"

4. In the online submission form you indicate that your data is not available for proprietary reasons and have provided a contact point for accessing this data. Please note that your current contact point is a co-author on this manuscript. According to our Data Policy, the contact point must not be an author on the manuscript and must be an institutional contact, ideally not an individual. Please revise your data statement to a non-author institutional point of contact, such as a data access or ethics committee, and send this to us via return email. Please also include contact information for the third party organization, and please include the full citation of where the data can be found.

5. We note that Figure 1 in your submission contain map/satellite images which may be copyrighted. All PLOS content is published under the Creative Commons Attribution License (CC BY 4.0), which means that the manuscript, images, and Supporting Information files will be freely available online, and any third party is permitted to access, download, copy, distribute, and use these materials in any way, even commercially, with proper attribution. For these reasons, we cannot publish previously copyrighted maps or satellite images created using proprietary data, such as Google software (Google Maps, Street View, and Earth). For more information, see our copyright guidelines: http://journals.plos.org/plosone/s/licenses-and-copyright.

6. Please ensure that you refer to Figure 1 in your text as, if accepted, production will need this reference to link the reader to the figure.

Additional Editor Comments:

Thank you for submission of this manuscript:

Causes and risk factors for an acute kidney injury outbreak among children in The Gambia, June – September 2022: A case-cohort study (PONE-D-24-22500)

Although the topic is interesting, there are several issues that must be corrected before further processing of this manuscript can be done.

Abstract

1. No study objective written

Introduction

1. Add more data and update about death toll number, fatality rate, what kind of medications, what substances cause this issue (DEG/EG) (all available here: https://www.kidney-international.org/article/S0085-2538(22)01025-0/fulltext)

2. No citation from line 69-79, please add more

3. Reference 5 is missing

Methods

1. No need “” in STROBE

2. Add primary data collection in addition to “the national epidemiological surveillance system, line-list and preliminary epidemiological reports”

3. Update DHS data to 2019-2020 (https://dhsprogram.com/publications/publication-FR369-DHS-Final-Reports.cfm) and write the interpretation

4. Add definition of suspected, probable, and confirmed AKI cases in addition to general AKI definition

5. Add proper citation for SAS as analytical software

6. Any explanation about controls who may become cases (no AKI is the inclusion of control)

7. Please rewrite: All 82 confirmed cases were sought for inclusion in the study (contradictive since the authors stated that they cannot include every case due to financial constraints)

8. Eleven patients from suspected and probable cases? Please explain from many cases of this group

9. It is vague: probability sampling technique (there are many specific techniques, such as cluster sampling), please correct it

10. Add the Gambia income determination (quantitative threshold such as household annual income and income per capita is more suitable, not only about pipe – add reference only if there is no reference to the quantitative determination of income)

11. Looks like there is more than 3:1 control:cases (nearer to 4:1)

Result

1. It is inconsistent, where is the 82 confirmed cases as mentioned earlier?

2. Quranic education is not a form of formal education, consider to merge with no education

3. What is the threshold for younger child? Two Gambian children with a one-year difference seems not standard

Discussion

1. Add citation in the first paragraph

2. Is there any specific measure that already done by the government (in addition to withdrawing the drug) – to prevent similar event in the future

Figure

1. From where is figure 1? If from other sources, state the permission to publish, while if created by the authors, please add about software utilized to create it

Table 1&2

1. Do statistical analysis to compare characteristics between each group. Descriptive statistics are not enough – update in the methods

Table 1

1. The authors wrote that they include all 82 patient; however, the one mentioned in the table is only 63 – please explain about this difference

2. Is all data have abnormal data distribution? Because ratio is mentioned in median (IQR), please check

3. Add MUAC long form

4. Why is there many missing weight and height (>60?)

5. Remove Relation caregiver – child (mother), n (%)

6. Remove Education mother (High school), n (%) 321 90 (34.9) 19 (30.2)

7. Consider removing: Education head of family, n (%)

Funding

1. Add funding number

Reviewers' comments:

Reviewer's Responses to Questions

**Comments to the Author**

1. Is the manuscript technically sound, and do the data support the conclusions?

Reviewer #1: Partly

Reviewer #2: Yes

Reviewer #3: Yes

Reviewer #4: Yes

2. Has the statistical analysis been performed appropriately and rigorously? 

Reviewer #1: No

Reviewer #2: I Don't Know

Reviewer #3: Yes

Reviewer #4: Yes

3. Have the authors made all data underlying the findings in their manuscript fully available?

Reviewer #1: Yes

Reviewer #2: Yes

Reviewer #3: Yes

Reviewer #4: No

4. Is the manuscript presented in an intelligible fashion and written in standard English?

Reviewer #1: Yes

Reviewer #2: Yes

Reviewer #3: Yes

Reviewer #4: Yes

5. Review Comments to the Author

Reviewer #1: Incidences like the one which happened in Gambia expose the perils of adulterated medicine and begs better regulatory oversight and control measures to prevent such incidences. Authors have done an epidemiological investigation to confirm the cause and contributory risk factors. I have following comments:

Material and methods need more clarity. Authors state that cases and controls were selected from the main cohort. Were all the cases of AKI in the region during the study period selected? Or was it studied in only the main cohort which as per the study is only 625 patients. How was the main cohort selected?

How did the authors ensure that all the cases of AKI were selected?

What was the study period? It needs to be mentioned.

What was the criteria for selecting sub cohort? Why was there a necessity to create a sub cohort?

The flowchart in figure 2 mentions that there were 11 patients who were unreachable and eliminated. Once again it mentions there were 5 more unreachable and number of suspected/probable cases have thus reduced to 6. This is confusing. Number of cases who were removed because of unreachable should be clubbed together only once.

AKI definition is not clear. There are clear cut definitions of AKI as per KDIGO. That should be mentioned.

Was any biochemical tests done? Like serum creatinine? If yes then please provide details

Please provide detailed table with logistic regression. What factors were adjusted for? There seems to be many confounders.

The tables need to be redone completely. Under AKI heading sum of so many columns is coming as 64 but the number mentioned is 63.

In Table 2 under the clinical characteristics the N value is varying everytime (eg. 319, 318,312…) rather than being constant at the number 321. The numbers are not adding up.

More number of patients who developed AKI had fever, vomiting. Was this before (as a part of disease process) or after giving the medicines?

Reviewer #2: This is a well written and relevant article. I would suggest a few modifications to the draft to make it useful to readers of the journal

Paragraph 1, line 3: Can include obstructive uropathy as other causes of AKI.

Line 4 - elaborate why burden of AKI is high in children in sub-saharan Africa

For secondary exposures - elaborate on concomitant drugs (antibiotics, antimalarial drugs),

toxins from animals and industrial toxins - which groups or products were analysed? Are they associated with AKI? Although the statistical analyses paragraph mentions natural and industrial toxins - no mention of it in results or analyses shared.

Did the authors attempt to examine the time between consumption of adulterated medicines and development of AKI? A similar interval between ingestion and onset of AKI in cases would be an important aspect to prove causation in all cases.

A breif description on the pathophysiology of AKI due to DEG in the discussion section would further improve the article.

Reviewer #3: The manuscript entitled “Causes and risk factors for an acute kidney injury outbreak among children in The Gambia, June – September 2022: A case-cohort study” discusses the consumption of diethylene glycol and ethylene glycol adulterated medicines manufactured by one pharmaceutical company, leading to acute kidney injury (AKI) in children eight years old or younger. Health providers and the public need to be sensitized about the dangers of adulterated medicines, and the government needs access to resources and expertise in pharmaceutical quality control.

Additionally, the definition of AKI should be corrected. It is not accurately defined as the serum creatinine level above normal. Instead, it would be best to use the pediatric AKI definitions such as p-RIFLE, AKIN, or KDIGO.

Reviewer #4: Overall: Bittaye et al report on a case-control study conducted in the Gambia in order to investigate an outbreak of AKI among young children in 2022. They found higher odds of developing AKI among children who had ingested adulterated medicines compared to children who did not.

The study addresses an important event. Concerns remain on how the authors dealt with recall bias especially that data was collected/ interviews were conducted months after the cases and that in the interim there had been countrywide sensitization of communities on the symptoms of illness and the suspected culprit medications? Perhaps a study limitation section could address this concern.

The authors also need to define better what a confirmed case is and what tests/criteria was used to conclude that AKI was of unknown etiology. Suggest the use of phrases “epidemiological” definition of AKI and “clinical” definition of AKI.

Introduction:

1. Line 66-67 “The primary causes of AKI include extrarenal pathology such as prerenal azotemia, specific kidney diseases such as acute interstitial nephritis or non-specific conditions such as nephrotoxicity”

- Generally, the term “primary cause” of AKI would imply conditions originating in the kidney. please us alternative phrase

- Post-renal causes also important cause in children

2. Line 67-68 “Drug induced nephrotoxicity is the third commonest cause of AKI.3”

Not in children. More common causes are infections such as sepsis, malaria, acute gastroenteritis, insect and snake bites, nephrotoxins such as haeme, congenital anomalies of the kidneys etc Kindly review papers below:

-Olowu WA, Niang A, Osafo C, et al. Outcomes of acute kidney injury in children and adults in 347 sub-Saharan Africa: a systematic review. The Lancet Global Health. 2016;4(4):e242-e250.

-Macedo E, Cerdá J, Hingorani S, Hou J, Bagga A, Burdmann EA, et al. Recognition and management of acute kidney injury in children: the ISN 0by25 Global Snapshot study. PLoS One. 2018; 13(5):e0196586. pmid:29715307

-Lameire N, Van Biesen W, Vanholder R. Epidemiology of acute kidney injury in children worldwide, including developing countries. Pediatr Nephrol. 2017; 32(8):1301–14. pmid:27307245

-Susantitaphong P, Cruz DN, Cerda J, Abulfaraj M, Alqahtani F, Koulouridis I, et al. World incidence of AKI: a meta-analysis. Clin J Am Soc Nephrol. 2013; 8(9):1482–93. pmid:23744003

3. line 69-71: “a pediatric nephrologist at Edward 70 Francis Small Teaching Hospital (ETSTH) – the only teaching hospital in The Gambia – observed an unusual increase in the number of cases of AKI among children from five months to seven years of age..”

- A little more detail on the usual numbers of AKI seen at the teaching hospital and how many were observed which caused the pediatric nephrologist to raise alarm ?

4. Line 76-78: “Nine samples of medicines taken by children with AKI and sent for toxicological tests were found to contain unacceptable levels of diethylene glycol and ethylene glycol”

Were these nine samples the actual ones retrieved from the AKI children or nine samples similar to ones (i.e. same brand, manufacturer) taken by the AKI children? Please clarify

5. line 74-75 “No toxic substances were detected in other pediatric medicinal products supplied 75 in The Gambia by other pharmaceutical companies.”

This sentence should come after line 78

6. Line 79-80: “Initial epidemiological investigations conducted by the MoH and US CDC were inconclusive as to the cause7”

- Some detail on why this report was inconclusive please?

- How does the current study address these reasons e.g the recall bias that was pointed out, the methods used in current study?

Methods:

1. Line 114-115: “or confirmed with serum creatinine above normal and of unknown etiology, diagnosed within the period June to September 2022.”

Please clarify the criteria for a “confirmed case”. i.e the level of increase above normal that would be considered as AKI.

“unknown etiology” – what tests were conducted on the cases to rule out the other AKI etiologies” ? I think a checklist of what was used to rule out usual causes would help.

2. Line 27 under sampling procedure:” All 82 confirmed cases were sought for inclusion in the study. We also sought to include eleven suspected or probable cases from WR1 and WR2.”

How were the cases identified? Examining hospital records? How many hospitals? Or was a national registry used.

3. Line 142 “The research team comprised surveillance officers that had interacted with the children previously.”

When previously? During the MOH/cdc study or when the outbreak was first identified?

Would this have been a risk for recall bias

Results:

1. Table 1 and 2, on which narration from line 187-201 is based, do not have p values to help show if the differences in the variables between the cases and controls were statistically significant

2. How many of the children were dialyzed?

3. Length of time between onset of anuria/oliguria and access to health facility

Discussion:

1. Line 228-229: “The temporal relationship between the importation of the suspected medicines into 229 the country, their consumption and development of AKI among the children is supportive of this hypothesis.”

This is being mentioned for the first time here. Needs to have a citation please or perhaps this should be expanded upon in the introduction section before being mentioned here.

2. Did dialysis access have any impact on the high mortality among the cases?

6. PLOS authors have the option to publish the peer review history of their article (what does this mean? ). If published, this will include your full peer review and any attached files.

**Do you want your identity to be public for this peer review?** For information about this choice, including consent withdrawal, please see our Privacy Policy .

Reviewer #1: **Yes: ** Pranaw Kumar Jha

Reviewer #2: **Yes: ** Sukanya Govindan

Reviewer #3: No

Reviewer #4: No

---

## [Author Response · Author response to Decision Letter 1]

26 Aug 2024

Comments Responses Page

Additional Editor Comments:

Thank you for submission of this manuscript:

Causes and risk factors for an acute kidney injury outbreak among children in The Gambia, June – September 2022: A case-cohort study (PONE-D-24-22500)

Although the topic is interesting, there are several issues that must be corrected before further processing of this manuscript can be done.

Abstract

1. No study objective written Thank you for pointing that out. We have provided an objective in both the abstract and at the end of the introduction Page 3, lines 44-45; Page 5, lines 96-98

Introduction

1. Add more data and update about death toll number, fatality rate, what kind of medications, what substances cause this issue (DEG/EG) (all available here: https://www.kidney-international.org/article/S0085-2538(22)01025-0/fulltext)

We have confirmed from the latest linelist as of 03rd of December 2022 that the number of confirmed deaths was 66. We can provide the linelist on request. As investigators on the ground, we attended Incident Management Team meetings headed by the MoH of the Gambia where the number of deaths was confirmed to be 66.

We have added information on the medications that was suspected to cause that issue. No changes made

Page 4, lines 74, 75

2. No citation from line 69-79, please add more Thank you for your comment. We have provided more citations Page 4, lines 71-81

3. Reference 5 is missing We have added the missing reference which is now reference number 9 Page 4, line 71

Methods

1. No need “” in STROBE We have removed the “” Page 5, lines 101, 102

2. Add primary data collection in addition to “the national epidemiological surveillance system, line-list and preliminary epidemiological reports” Thank you for your comments: We have reviewed as such: “We collected primary data and reviewed data from the national epidemiological surveillance system, line-list and preliminary epidemiological reports” Page 5, line 109; Page 6, lines 110-11

3. Update DHS data to 2019-2020 (https://dhsprogram.com/publications/publication-FR369-DHS-Final-Reports.cfm) and write the interpretation We have updated the data source and provided the reference Page 7, lines 120, 121

4. Add definition of suspected, probable, and confirmed AKI cases in addition to general AKI definition We have added definitions as suggested Page 6, lines 132-139

5. Add proper citation for SAS as analytical software We have added a proper citation for the SAS software Page 7, line 146

6. Any explanation about controls who may become cases (no AKI is the inclusion of control) Thank you for your comment. In this study no control became a case. We accounted for it in the sample size calculation in case it happened, but it did not happen. We have specified that under the Participants section in the results Page 7, line 147

7. Please rewrite: All 82 confirmed cases were sought for inclusion in the study (contradictive since the authors stated that they cannot include every case due to financial constraints) Thank you for your comment. We have reformulated the sentence and emphasized that the final constraints referred to the inclusion of suspected/probable cases from other regions but did not refer to confirmed cases Page 7, line 138-140

8. Eleven patients from suspected and probable cases? Please explain from many cases of this group Thank you so very much for the comment. We have revised the sentence to clarify. WR1 and WR2 produced a total of eleven suspected/probable cases who were all included in the study. Hence, no sampling was done. Page 8, lines 152-155.

9. It is vague: probability sampling technique (there are many specific techniques, such as cluster sampling), please correct it We have specified the sampling technique used. Controls we selected using a simple random sampling technique Page 7, line 155

10. Add the Gambia income determination (quantitative threshold such as household annual income and income per capita is more suitable, not only about pipe – add reference only if there is no reference to the quantitative determination of income) Thank you for the very important comment. We provided details on the determination of the income and the methodology we used. Page 9, lines 185-196

11. Looks like there is more than 3:1 control:cases (nearer to 4:1) We enrolled all the controls planned for but some of the cases declined to participate in the study. Hence, the distortion in the ratio of cases to controls.

This does not affect the power of our study.

Result

1. It is inconsistent, where is the 82 confirmed cases as mentioned earlier? Thank you for your comment. We recruited 82 confirmed cases. Of these, we enrolled 58 and analysed 57 together with 6 suspected and probable cases to make a total of 63 cases. Page 11, Flow chart

2. Quranic education is not a form of formal education, consider to merge with no education For the inferential analysis, the education variable was categorised into binary; High and Low. Primary, Quranic and None formed the low category. According to the literature, health literacy is congruent with secondary and higher education.

3. What is the threshold for younger child? Two Gambian children with a one-year difference seems not standard Thank you for your comment. There is no threshold for younger child, The variable “Age” was considered as continuous (in year) in the analyses. The models compared two individuals with X and (X+1) as ages respectively. The younger child here will be the one with X year old.

Discussion

1. Add citation in the first paragraph We added some citations as suggested Page 16, line 260, 265

2. Is there any specific measure that already done by the government (in addition to withdrawing the drug) – to prevent similar event in the future Thank you for the comment. The products of the Company (Maiden pharmaceuticals Limited) were banned to be imported in the Gambia and the company closed. Internally, the government increased the control at the port of entry in the Gambia. Included in the introduction. Page 4, lines 86-88

Figure

1. From where is figure 1? If from other sources, state the permission to publish, while if created by the authors, please add about software utilized to create it We have provided the source of the figure 1 and the software used Page 6, line 117

Table 1&2

1. Do statistical analysis to compare characteristics between each group. Descriptive statistics are not enough – update in the methods Since we were describing groups, we did not provide p-values as per guidelines such as the STROBE and the CONSORT. It is recommended to use literature review and unbalanced variables identified

Table 1

1. The authors wrote that they include all 82 patients; however, the one mentioned in the table is only 63 – please explain about this difference Refer to our response to the comment 1 under Result:

Thank you for your comment. We recruited 82 confirmed cases. Of these, we enrolled 58 and analysed 57 together with 6 suspected and probable cases to make a total of 63 cases Page 11, Flow chart

2. Is all data have abnormal data distribution? Because ratio is mentioned in median (IQR), please check Yes, we checked, and the distribution of the continuous variables was not normal. So, we decided to present the median (IQR). Pages 11,12, table 1

3. Add MUAC long form We provided the long form of MUAC as mid-upper arm circumference Pages 12, table 1

4. Why is there many missing weight and height (>60?) Most of the missing data are from the dead cases that we did have the opportunity to measure. We used records to collect these variables which were missing in many of the records

5. Remove Relation caregiver – child (mother), n (%) We have renamed it as it refers to mother as caregiver Page 12, table 1

6. Remove Education mother (High school), n (%) 321 90 (34.9) 19 (30.2) We have removed it from the table Page 12, table 1

7. Consider removing: Education head of family, n (%) We have removed it from the table Page 12, table 1

Funding

1. Add funding number We have provided funding number as suggested Page 20, line 361

5. Review Comments to the Author

Reviewer #1: Incidences like the one which happened in Gambia expose the perils of adulterated medicine and begs better regulatory oversight and control measures to prevent such incidences. Authors have done an epidemiological investigation to confirm the cause and contributory risk factors. I have following comments:

Material and methods need more clarity.

Authors state that cases and controls were selected from the main cohort. Were all the cases of AKI in the region during the study period selected? Or was it studied in only the main cohort which as per the study is only 625 patients. How was the main cohort selected? Thank you for your comment. Yes, all the cases in the regions were selected.

The main cohort comprised all the children that resided in the villages from which the cases arose.

The sub-cohort was as a fraction of the main cohort determined by proportionality to the number of confirmed cases in a given village. Page 5, lines 11-102

How did the authors ensure that all the cases of AKI were selected? We worked from a line-list identified and confirmed by the Ministry of Health. No change

What was the study period? It needs to be mentioned. From 15 to 22 December 2022. Page 5, line 107

What was the criteria for selecting sub cohort? Why was there a necessity to create a sub cohort? There were no criteria to select the sub-cohort. The sub-cohort was a fraction of the main cohort that was enough to provide the number of controls needed.

The advantage of the sub-cohort was to ensure cost-effectiveness of the study while ensuring its scientific validity. Using the main cohort would have been very expensive with no extra advantage.

The flowchart in figure 2 mentions that there were 11 patients who were unreachable and eliminated. Once again it mentions there were 5 more unreachable and number of suspected/probable cases have thus reduced to 6.

This is confusing. Number of cases who were removed because of unreachable should be clubbed together only once. Thank you for your comment. Lumping them together does not give an accurate picture of the points at which the cases were not reachable. Hence, the decision to separate them in the flow chart.

AKI definition is not clear. There are clear cut definitions of AKI as per KDIGO. That should be mentioned. We have added definitions as suggested Page 6, lines 122-131

Was any biochemical tests done? Like serum creatinine? If yes, then please provide details Yes, biochemical tests were done namely serum creatinine and urea. But it was serum creatinine that was used in the case definition. An AKI case was diagnosed if the measured creatinine was 1.5-2 times greater than the reference Page 7, line 133

Please provide detailed table with logistic regression. What factors were adjusted for? There seems to be many confounders. Thank you so very much for your comment. The logistic regression was used to identify risk factors of AKI. In the identification of risk factors, by design, we do not look for the effect of an exposure on an outcome. Hence, there is nothing like adjusting for confounders.

The tables need to be redone completely. Under AKI heading sum of so many columns is coming as 64 but the number mentioned is 63. Thank you for your comment. It is correct. We have redone the table as suggested and provided actual number and percentage. Pages 11-12, table 1

In Table 2 under the clinical characteristics the N value is varying everytime (e.g. 319, 318,312…) rather than being constant at the number 321. The numbers are not adding up Thank you so very much for the comment. The table 2 was corrected as well. We have harmonised the totals Pages 13-14, Table 2

More number of patients who developed AKI had fever, vomiting. Was this before (as a part of disease process) or after giving the medicines? Yes, these symptoms were developed after exposure to the medicines

Reviewer #2: This is a well written and relevant article. I would suggest a few modifications to the draft to make it useful to readers of the journal

Paragraph 1, line 3: Can include obstructive uropathy as other causes of AKI. We have added as suggested Page 4, line 65

Line 4 - elaborate why burden of AKI is high in children in sub-saharan Africa We have elaborated as requested Page 4, lines 70-71

For secondary exposures - elaborate on concomitant drugs (antibiotics, antimalarial drugs), toxins from animals and industrial toxins - which groups or products were analysed?

Are they associated with AKI? Although the statistical analyses paragraph mentions natural and industrial toxins - no mention of it in results or analyses shared According to the literature, these are potential causes of AKI. So, we included them in the study assessed by self-report. We did not analyse any product as secondary exposures.

All these variables were included in the treatment model that was used to produce the IPTW for the MSM. The MSM does not assess for associations. The focus for the MSM was the main exposure.

However, using a multivariable logistic regression to identify risk factors, all these variables were entered into the models, but were dropped leaving concomitant drugs that we have shown in the results.

Page 16, table 3.

Did the authors attempt to examine the time between consumption of adulterated medicines and development of AKI? A similar interval between ingestion and onset of AKI in cases would be an important aspect to prove causation in all cases. Thank you for your comment. Yes, we have examined the temporary relationship between the exposure and the outcome and ensured the anteriority. We did not capture the exact number of days between consumption and development of symptoms. However, this was captured in the causality assessment at the same time as our study. Page 17, lines 263-265

A brief description on the pathophysiology of AKI due to DEG in the discussion section would further improve the article. We have added a brief description as suggested Page 16, lines 268-272

Reviewer #3: The manuscript entitled “Causes and risk factors for an acute kidney injury outbreak among children in The Gambia, June – September 2022: A case-cohort study” discusses the consumption of diethylene glycol and ethylene glycol adulterated medicines manufactured by one pharmaceutical company, leading to acute kidney injury (AKI) in children eight years old or younger. Health providers and the public need to be sensitized about the dangers of adulterated medicines, and the government needs access to resources and expertise in pharmaceutical quality control.

Additionally, the definition of AKI should be corrected. It is not accurately defined as the serum creatinine level above normal. Instead, it would be best to use the pediatric AKI definitions such as p-RIFLE, AKIN, or KDIGO. Regarding the recommendations “Health providers and the public need to be sensitized about the dangers of adulterated medicines, and the government needs access to resources and expertise in pharmaceutical quality control” was already provided under recommendations section.

We have elaborated on the case definitions. Page 20, lines 359 - 362

Page 7, lines 132-140

Reviewer #4: Overall: Bittaye et al report on a case-control study conducted in the Gambia in order to investigate an outbreak of AKI among young children in 2022. They found higher odds of developing AKI among children who had ingested adulterated medicines compared to children who did not.

The study addresses an important event. Concerns remain on how the authors dealt with recall bias especially that data was collected/ interviews were conducted months after the cases and that in the interim there had been countrywide sensitization of communities on the symptoms of illness and the suspected culprit medications? Perhaps a s

---

## [Decision Letter · Decision Letter 1]

PONE-D-24-22500R1Causes and risk factors for an acute kidney injury outbreak among children in The Gambia, June – September 2022: A case-cohort studyPLOS ONE

Dear Dr. Office,

Thank you for submitting your manuscript to PLOS ONE. After careful consideration, we feel that it has merit but does not fully meet PLOS ONE’s publication criteria as it currently stands. Therefore, we invite you to submit a revised version of the manuscript that addresses the points raised during the review process.

We look forward to receiving your revised manuscript.

Kind regards,

Tungki Pratama Umar, M.D.

Academic Editor

PLOS ONE

**Editor Comments** :

Thank you for your effort to revise this manuscript following four reviewers and my comments. From the evaluation, most of the comments have been resolved, however, there are several concerns remained in this manuscript, both from the reviewers and I, which made a revise decision to be selected. Here are my comments, please also check all reviewers' comments.

- In the objective, add "in Gambia" since the risk factors is limited in this country

- After we collected primary data, add by what method (e.g. interview)

- Software used to generate Gambia map figure has not been written, also I think the authors deleted the caption (as shown by track changes). Or the authors plan to remove this figure?

Reviewers' comments:

Reviewer's Responses to Questions

**Comments to the Author**

1. If the authors have adequately addressed your comments raised in a previous round of review and you feel that this manuscript is now acceptable for publication, you may indicate that here to bypass the “Comments to the Author” section, enter your conflict of interest statement in the “Confidential to Editor” section, and submit your "Accept" recommendation.

Reviewer #1: (No Response)

Reviewer #2: All comments have been addressed

Reviewer #3: All comments have been addressed

Reviewer #4: (No Response)

2. Is the manuscript technically sound, and do the data support the conclusions?

Reviewer #1: Partly

Reviewer #2: Yes

Reviewer #3: Yes

Reviewer #4: Yes

3. Has the statistical analysis been performed appropriately and rigorously? 

Reviewer #1: No

Reviewer #2: I Don't Know

Reviewer #3: Yes

Reviewer #4: Yes

4. Have the authors made all data underlying the findings in their manuscript fully available?

Reviewer #1: Yes

Reviewer #2: Yes

Reviewer #3: Yes

Reviewer #4: No

5. Is the manuscript presented in an intelligible fashion and written in standard English?

Reviewer #1: Yes

Reviewer #2: Yes

Reviewer #3: Yes

Reviewer #4: Yes

6. Review Comments to the Author

Reviewer #1: Thanks for the rebuttal. I have following comments:

Also, authors should provide the p-value for baseline sociodemographic characteristics to show whether any of the baseline characteristics were different between the two groups.

Please mention what was the criteria for selecting factors for logistic regression analysis.

“In the identification of risk factors, by design, we do not look for the effect of an exposure on an outcome. Hence, there is nothing like adjusting for confounders.” The rebuttal is not right. The cause and effect relationship will be meaningful only if one adjusts for the baseline confounders which are very well present in the study. The groups developing AKI also had higher use of anti-inflammatory drugs, antibiotics, acetaminophen. How do we know that these were not the causes of the AKI rather than the suspected adulterated cough syrups. To say that the adulterated drug was the culprit authors should adjust for these factors in the study. Authors are not looking for just risk factors. They are putting a causal association with adulterated drug and hence it becomes important. “The acute kidney injury outbreak that occurred among children in The Gambia in the period June through September 2022 was causally associated with the ingestion of adulterated medicines.”

Reviewer #2: The revised version of the article addresses the queries posted by me and I suggest that the article be accepted for publication.

Reviewer #3: It would be interesting to explain the categorical variable "primary water source for drinking" to better understand the differences related to its score with 3 ordinal variables.

Reviewer #4: The manuscript reads very well.

Methods:

1. For the definition of controls: Were these all well children from the community from which the AKI cases originated or were some of them unwell with some other sickness at the time of the outbreak? Kindly clarify this in the definition of controls.

2. Line 129-130: “fever, vomiting, 130 diarrhea, cough, with history of syrup consumption or a child 8 years old”

This was used as one of the three definitions of a suspected AKI case.

This is too imprecise a definition for AKI since no renal specific symptoms are included in this definition. The mere fact that a child consumed syrup would be too imprecise to label them as suspected cases.

How many subjects were included because of this particular definition?

If subjects recruited on the basis of this definition did not have AKI, their inclusion could skew the results.

[ Additionally from table 1/2 some of the controls also had vomiting, diarrhoea, cough and had ingested adulterated medications suggesting that the presence of renal specific symptoms in the definition is important]

Discussion:

1. Line 272-274 “The alternative explanation of E. coli infection is unlikely because only 8% of cases presented with bloody 273 diarrhea which increases the risk of developing serious complications such as acute renal failure. Only two 274 samples tested positive for Shiga toxin-producing E. coli.”

It would be good to state if the clinical picture (low haemoglobin, low platelets, schistocytes) was present or not in any of the subjects if this data is available.

2. Dialysis was unavailable and this contributed to the high mortality rates.

Under recommendations, would be good to add a note that Africa needs heightened advocacy and provision of organ support services in general and paediatric dialysis facilities in particular.

7. PLOS authors have the option to publish the peer review history of their article (what does this mean? ). If published, this will include your full peer review and any attached files.

**Do you want your identity to be public for this peer review?** For information about this choice, including consent withdrawal, please see our Privacy Policy .

Reviewer #1: **Yes: ** PRANAW KUMAR JHA

Reviewer #2: **Yes: ** Sukanya Govindan

Reviewer #3: No

Reviewer #4: No

---

## [Author Response · Author response to Decision Letter 2]

1 Nov 2024

RESPONSES TO THE REVIEWERS

PONE-D-24-22500R1

Causes and risk factors for an acute kidney injury outbreak among children in The Gambia, June – September 2022: A case-cohort study

PLOS ONE

Dear Dr. Office,

Thank you for submitting your manuscript to PLOS ONE. After careful consideration, we feel that it has merit but does not fully meet PLOS ONE’s publication criteria as it currently stands. Therefore, we invite you to submit a revised version of the manuscript that addresses the points raised during the review process.

We look forward to receiving your revised manuscript.

Kind regards,

Tungki Pratama Umar, M.D.

Academic Editor

PLOS ONE

Editor Comments:

Thank you for your effort to revise this manuscript following four reviewers and my comments. From the evaluation, most of the comments have been resolved, however, there are several concerns remained in this manuscript, both from the reviewers and I, which made a revise decision to be selected. Here are my comments, please also check all reviewers' comments.

- In the objective, add "in Gambia" since the risk factors is limited in this country

- After we collected primary data, add by what method (e.g. interview)

- Software used to generate Gambia map figure has not been written, also I think the authors deleted the caption (as shown by track changes). Or the authors plan to remove this figure?

Reviewers' comments:

Reviewer's Responses to Questions

Comments to the Author

1. If the authors have adequately addressed your comments raised in a previous round of review and you feel that this manuscript is now acceptable for publication, you may indicate that here to bypass the “Comments to the Author” section, enter your conflict of interest statement in the “Confidential to Editor” section, and submit your "Accept" recommendation.

Reviewer #1: (No Response)

Reviewer #2: All comments have been addressed

Reviewer #3: All comments have been addressed

Reviewer #4: (No Response)

2. Is the manuscript technically sound, and do the data support the conclusions?

Reviewer #1: Partly

Reviewer #2: Yes

Reviewer #3: Yes

Reviewer #4: Yes

3. Has the statistical analysis been performed appropriately and rigorously?

Reviewer #1: No

Reviewer #2: I Don't Know

Reviewer #3: Yes

Reviewer #4: Yes

4. Have the authors made all data underlying the findings in their manuscript fully available?

Reviewer #1: Yes

Reviewer #2: Yes

Reviewer #3: Yes

Reviewer #4: No

5. Is the manuscript presented in an intelligible fashion and written in standard English?

Reviewer #1: Yes

Reviewer #2: Yes

Reviewer #3: Yes

Reviewer #4: Yes

Comments Response Pages, lines

1 Reviewer #1: Thanks for the rebuttal. I have following comments:

Also, authors should provide the p-value for baseline sociodemographic characteristics to show whether any of the baseline characteristics were different between the two groups.

Please mention what was the criteria for selecting factors for logistic regression analysis.

“In the identification of risk factors, by design, we do not look for the effect of an exposure on an outcome. Hence, there is nothing like adjusting for confounders.” The rebuttal is not right. The cause-and-effect relationship will be meaningful only if one adjusts for the baseline confounders which are very well present in the study. The groups developing AKI also had higher use of anti-inflammatory drugs, antibiotics, acetaminophen. How do we know that these were not the causes of the AKI rather than the suspected adulterated cough syrups. To say that the adulterated drug was the culprit authors should adjust for these factors in the study. Authors are not looking for just risk factors. They are putting a causal association with adulterated drug and hence it becomes important. “The acute kidney injury outbreak that occurred among children in The Gambia in the period June through September 2022 was causally associated with the ingestion of adulterated medicines.”

Thank you for your comment. The practice of providing p-values for comparing baseline sociodemographic characteristics to determine differences between groups is considered outdated, as evidenced by the references below. Use of p-values to compare baseline characteristics is not appropriate because:

1. There is no hypothesis to test regarding these baseline characteristics.

2. Inferential measures such as standard errors and confidence intervals should not be used to describe the variability of characteristics, and significance tests should be avoided in descriptive tables.

(Vandenbroucke, Jan P. et al., 2007. For the STROBE Initiative. Strengthening the Reporting of Observational Studies in Epidemiology (STROBE): Explanation and Elaboration. Epidemiology 18(6):p 805-835, November 2007. | DOI: 10.1097/EDE.0b013e3181577511

Dales LG, Ury HK. An improper use of statistical significance testing in studying covariables. International Journal of Epidemiology. 1978;7:373–5)

Thank you for your comment. We used logistic regression for the second objective to identify risk factors of the AKI outbreak.

We used the backward logistic regression model to identify risk factors for the AKI outbreak. This procedure is good for exploratory data analysis to identify the most influential variables when there is little prior knowledge about the data. This was the case for this outbreak. Backward logistic regression helps to systematically remove variables that do not significantly contribute to the model, which includes confounders.

We used a conservative cut-off p-value of ≤ 0.20 for elimination of non-influential variables and used a cut-off p-value of ≤ 0.05 to determine the final risk factors.

By doing so, the procedure retained only the most relevant predictors thereby reducing the potential bias introduced by confounders.

Important to note that for this objective, we were not assessing for a causal association but just identifying factors that are associated with the outcome. The causal association of ingestion of adulterated medicines (main exposure) with AKI was determined using the MSM which by design controls for confounding.

No change in the text

No change in the text

Reviewer #4: The manuscript reads very well.

Methods:

1. For the definition of controls: Were these all well children from the community from which the AKI cases originated or were some of them unwell with some other sickness at the time of the outbreak? Kindly clarify this in the definition of controls.

2. Line 129-130: “fever, vomiting, 130 diarrhea, cough, with history of syrup consumption or a child 8 years old”

This was used as one of the three definitions of a suspected AKI case.

This is too imprecise a definition for AKI since no renal specific symptoms are included in this definition. The mere fact that a child consumed syrup would be too imprecise to label them as suspected cases.

How many subjects were included because of this particular definition?

If subjects recruited on the basis of this definition did not have AKI, their inclusion could skew the results.

[ Additionally from table 1/2 some of the controls also had vomiting, diarrhoea, cough and had ingested adulterated medications suggesting that the presence of renal specific symptoms in the definition is important]

The main criterion for being a control was that the child did not have AKI. However, we went ahead to assess for the presence of other illnesses that the controls had at the time when the event happened. This assessment was done for both cases and controls. So, our definition for the control remains the same.

Thank you for your observation. We agree that the first definition is very imprecise. This was the first definition and was later refined to the latter two. Most of the cases (49; 78%) were assessed using the criteria with renal specific symptoms (anuria/oliguria). Only 14 cases were assessed using the simple definition. However, they were confirmed by a nephrologist to have had AKI using other clinical criteria.

No change in the text

No change in the text

Reviewer #4:

Discussion:

1. Line 272-274 “The alternative explanation of E. coli infection is unlikely because only 8% of cases presented with bloody 273 diarrhea which increases the risk of developing serious complications such as acute renal failure. Only two 274 samples tested positive for Shiga toxin-producing E. coli.”

It would be good to state if the clinical picture (low haemoglobin, low platelets, schistocytes) was present or not in any of the subjects if this data is available.

2. Dialysis was unavailable and this contributed to the high mortality rates.

Under recommendations, would be good to add a note that Africa needs heightened advocacy and provision of organ support services in general and paediatric dialysis facilities in particular.

We agree with you on the need to add more clinical and laboratory characteristics. Unfortunately, these data were not collected and therefore not available since the study was done retrospectively.

Thank you for the suggestion. This has been incorporated.

No change in the text

Page 20, lines 368 - 369

---

## [Decision Letter · Decision Letter 2]

PONE-D-24-22500R2Causes and risk factors for an acute kidney injury outbreak among children in The Gambia, June – September 2022: A case-cohort studyPLOS ONE

Dear Dr. OKEIBUNOR,

Thank you for submitting your manuscript to PLOS ONE. After careful consideration, we feel that it has merit but does not fully meet PLOS ONE’s publication criteria as it currently stands. Therefore, we invite you to submit a revised version of the manuscript that addresses the points raised during the review process.

We look forward to receiving your revised manuscript.

Kind regards,

Tungki Pratama Umar, M.D.

Academic Editor

PLOS ONE

Journal Requirements:

**Additional Editor Comments:**

Thank you for your effort to revise this manuscript following three reviewers' comments. However, I noted that the response to the editor comments are inadequate (not listed). Thus, please recheck my comment as also mentioned earlier.

- After we collected primary data, add by what method (e.g. interview)

- Software used to generate Gambia map figure has not been written, also I think the authors deleted the caption (as shown by track changes). Or the authors plan to remove this figure?

Reviewers' comments:

Reviewer's Responses to Questions

**Comments to the Author**

1. If the authors have adequately addressed your comments raised in a previous round of review and you feel that this manuscript is now acceptable for publication, you may indicate that here to bypass the “Comments to the Author” section, enter your conflict of interest statement in the “Confidential to Editor” section, and submit your "Accept" recommendation.

Reviewer #1: All comments have been addressed

Reviewer #3: All comments have been addressed

Reviewer #4: All comments have been addressed

2. Is the manuscript technically sound, and do the data support the conclusions?

Reviewer #1: Yes

Reviewer #3: Yes

Reviewer #4: Yes

3. Has the statistical analysis been performed appropriately and rigorously? 

Reviewer #1: Yes

Reviewer #3: Yes

Reviewer #4: Yes

4. Have the authors made all data underlying the findings in their manuscript fully available?

Reviewer #1: Yes

Reviewer #3: Yes

Reviewer #4: No

5. Is the manuscript presented in an intelligible fashion and written in standard English?

Reviewer #1: Yes

Reviewer #3: Yes

Reviewer #4: Yes

6. Review Comments to the Author

Reviewer #1: (No Response)

Reviewer #3: The manuscript is very interesting. I have the following comment: references 4 and 8 are the same. Please review the numbering of the corrected references in the article.

Reviewer #4: "Only 14 cases were assessed using the simple definition. However,

they were confirmed by a nephrologist to have had AKI using other clinical criteria."

Kindly provide a note in the manusript stating that the first definition was later ammended

7. PLOS authors have the option to publish the peer review history of their article (what does this mean? ). If published, this will include your full peer review and any attached files.

**Do you want your identity to be public for this peer review?** For information about this choice, including consent withdrawal, please see our Privacy Policy .

Reviewer #1: **Yes: ** Pranaw Kumar Jha

Reviewer #3: No

Reviewer #4: No

---

## [Author Response · Author response to Decision Letter 3]

14 Apr 2025

RESPONSES TO THE REVIEWERS

PONE-D-24-22500R2

Causes and risk factors for an acute kidney injury outbreak among children in The Gambia, June – September 2022: A case-cohort study.

PLOS ONE

Comments Response Pages, lines

Additional Editor Comments:

Thank you for your effort to revise this manuscript following three reviewers' comments. However, I noted that the response to the editor comments are inadequate (not listed). Thus, please recheck my comment as also mentioned earlier.

- After we collected primary data, add by what method (e.g. interview)

Thank you for your comments. We collected primary data using interviewer-administered questionnaires and reviewed data from the national epidemiological surveillance system, line-list and preliminary epidemiological reports. Line 109-110

- Software used to generate Gambia map figures has not been written, also I think the authors deleted the caption (as shown by track changes). Or the authors plan to remove this figure? Thank you so very much for your remarks. In our previous versions, the figure was deleted by mistake. We addressed it by bringing it back Lines 116 – 117, Page 6

6. Review Comments to the Author

Reviewer #3: The manuscript is very interesting. I have the following comment: references 4 and 8 are the same. Please review the numbering of the corrected references in the article.

Thank you so very much for the remark. We removed the double reference, checked for the accuracy of the references, and renumbered them appropriately Lines 71 to 443; Pages 4 to 22

Reviewer #4: "Only 14 cases were assessed using the simple definition. However, they were confirmed by a nephrologist to have had AKI using other clinical criteria."

Kindly provide a note in the manuscript stating that the first definition was later amended.

Thank you for your comment. We provided a sentence in the manuscript to address the comment. “The definition of a suspected case was revised to enhance its sensitivity by incorporating symptoms specific to renal conditions.” Lines 135 to 136;

---

## [Editor Report · Decision Letter 3]

Causes and risk factors for an acute kidney injury outbreak among children in The Gambia, June – September 2022: A case-cohort study

PONE-D-24-22500R3

Dear Dr. OKEIBUNOR,

We’re pleased to inform you that your manuscript has been judged scientifically suitable for publication and will be formally accepted for publication once it meets all outstanding technical requirements.

Kind regards,

Tungki Pratama Umar, M.D.

Academic Editor

PLOS ONE
---

## [Editor Report · Acceptance letter]

PONE-D-24-22500R3

PLOS ONE

Dear Dr. OKEIBUNOR,

I'm pleased to inform you that your manuscript has been deemed suitable for publication in PLOS ONE. Congratulations! Your manuscript is now being handed over to our production team.

Kind regards,

on behalf of

Dr. Tungki Pratama Umar

Academic Editor

PLOS ONE